# Short-Time Fatigue Life Estimation for Heat Treated Low Carbon Steels by Applying Electrical Resistance and Magnetic Barkhausen Noise

**DOI:** 10.3390/ma16010032

**Published:** 2022-12-21

**Authors:** Haoran Wu, Jonas Anton Ziman, Srinivasa Raghavan Raghuraman, Jan-Erik Nebel, Fabian Weber, Peter Starke

**Affiliations:** 1Department of Materials Science and Materials Testing, Institute QM3, University of Applied Sciences Kaiserslautern, D-67659 Kaiserslautern, Germany; 2Faculty of Natural Sciences and Technology, Saarland University, D-66123 Saarbruecken, Germany

**Keywords:** electrical resistance, magnetic Barkhausen noise, NDT, fatigue life, constant amplitude test, load increase test, tensile test, load-free sequence, low carbon steel, MaRePLife, StressLife

## Abstract

Tensile tests and fatigue tests on differently heat-treated low carbon (non- and low-alloy) steels were conducted and accompanied by non-destructive electrical resistometric (ER) and magnetic Barkhausen noise (MBN) measuring devices, in order to establish an improved short-time fatigue life estimation method according to StressLife. MaRePLife (**Ma**terial **Re**sponse **P**artitioning) is the hereby proposed method for calculating S–N curves in the HCF regime, based on the partitioning of material responses acquired during the above-mentioned mechanical tests. The rules were set to make use of the information gathered from pre-conducted tensile tests, which helps to determine the parameters of two load increase tests (LIT) and two constant amplitude tests (CAT). The results of the calculated S–N curves were satisfactory and could be verified by more separately conducted fatigue tests on specimens under different material conditions.

## 1. Introduction

For stress/force-controlled fatigue tests, a power law relationship according to Basquin is usually applied to describe S–N curves mathematically [1]. However, for strain-controlled tests, the Coffin–Manson relationship is used to describe the low-cycle fatigue (LCF) regime [2,3], while in the high-cycle fatigue (HCF) regime, the total strain–life curve (*ε_a,t_*-*N_f_*) is separated into a plastic portion (*ε_a,p_*), described via the Coffin–Manson relationship, and an elastic portion (*ε_a,e_*), related to Basquin’s equation. The conventional way of gathering S–N data is time- and cost-consuming; even for the HCF regime, at least 15 tests are required. Therefore, methods of accelerating this process or reducing the costs are always of interest. Researchers have paid attention to integrate non-destructive testing (NDT) methods into fatigue tests, so that more physical quantities can be measured and further processed for obtaining additional information regarding the fatigue processes compared to conventional strain, stress or the number of cycles to failure.

For example, electrical resistance (ER) has been researched for a long time after the pioneering work of Matthiessen [4]. The relation between mechanical stress and ER has been constantly researched ever since [5,6,7,8,9,10,11,12,13], even in this decade [14,15,16]. ER has also been used for investigating the fatigue properties of conductive materials [17,18,19,20,21,22,23,24,25]. Information regarding the defect density in bulk materials can be obtained via electrical resistance measurements [26,27], which can explain some early-stage fatigue phenomena.

In addition to ER, micromagnetic-based methods can also be used in this context, such as eddy current testing, incremental permeability, Fourier analysis of the tangential magnetic field, and magnetic Barkhausen noise (MBN) [27]. More papers in this field regarding the MBN technique [28,29,30,31,32,33,34,35,36] has already been introduced in [37].

In this work, the authors propose a new method named MaRePLife (**Ma**terial **Re**sponse **P**artitioning), derived from StressLife [38], aiming at calculatng S–N curves of differently heat-treated low carbon steels based on NDT measurands. The features used here are the change in the MBN signal ratio Δ*φ_MBN_* and the change in the ER ratio Δ*φ_R_*. This new approach proposes a way to define the experimental parameters clearly, so that the procedure can be generally fixed and applied for random metallic materials. Results of the calculated S–N curves are shown and discussed. In order to facilitate reading, explanations regarding the symbols and abbreviations in this work are listed in Table 1.

## 2. Materials and Methods

### 2.1. Material and Machining of Specimens

The tests were carried out on specimens of SAE 1020 (1.1149, C22R) and SAE 5120 (1.7149, 20MnCrS5) steels. The chemical compositions for both materials are summarized in Table 2 according to the manufacturer’s specifications and our own measurements, along with the specified limit values according to DIN EN 10083-2 and DIN EN 10084.

Within the framework of the research project DFG STA 1133/10, the specimens were tested in normalized conditions (“+N”, SAE 1020: *T_aust._* = 860 °C, SAE 5120: *T_aust._* = 840 °C) and in two quenched and tempered conditions (“+QT250” and “+QT650” with *T_temp._* = 250 °C and 650 °C, respectively, *t_temp._* = 60 min). Micrographs of both steels under different conditions, taken by an OLYMPUS DSX 1000 (EVIDENT Europe GmbH, Hamburg, DE, Germany) digital microscope, are given in Figure 1.

Both micrographs of SAE 1020 and SAE 5120 in normalized conditions (Figure 1a,b) show an obvious ferritic–pearlitic microstructure. In these micrographs, ferrite is displayed as whitish areas and pearlite is indicated by black cementite lamellas and white ferrite areas in between. For SAE 5120, the grain size of the ferrite is only up to approx. 30 µm, while the grain size of the ferrite in SAE 1020 is much larger, on average up to 50 µm. For all the specimens quenched and tempered (Figure 1c–f), typical lath-shaped martensite phases can be seen. The ferritic regions are irregularly arranged in blocks or strips between the bainite/martensitic slats and are easier to be identified if the tempering temperature is higher. The larger amount of ferrite is, therefore, an indication of a higher recovery rate of the meta-stable martensite. Due to the higher Mn content in SAE 5120, the martensitic transformation temperature is lower, so more retained austenite can be observed. The carbide phases in Figure 1f are, therefore, also better distributed in comparison to SAE 1020 +QT250, as shown in Figure 1e.

The geometries of the applied tensile and fatigue specimen are given in Figure 2.

### 2.2. Test Setup

Tensile tests were conducted on a Zwick Roell (ZwickRoell GmbH & Co., KG, Ulm, Germany) test system type 14,750 with *F_max_* = 100 kN. It utilizes a tactile sensor arm extensometer, which is attached directly to the specimen via knife edges mounted on the sensor arms according to the pre-programmed test procedure. The measuring length *L_0_* was set to 50 mm. The system was also modified with additional measurement devices, as shown in Figure 3, in order to fulfil the purposes of this research project. For obtaining the tensile strength *R_m_* and the yield strength *R_p0.2_*, tensile tests for each material condition were conducted conventionally according to the method A in DIN EN ISO 6892-1 without any strain-holding sequences at first, so that the stress–strain curve is smooth, and the values can be read out by the software easily. To characterize the MBN response during tensile tests, strain-holding sequences were inserted, during which the strain was held constant for a certain duration, so that a possible temperature increase due to plastic deformation could be compensated for before the start of the MBN measurement. At the end of the holding sequences, the MBN sensor was slid to the specimen manually and held hand-tight against the specimen surface during the MBN measurement. Then, the sensor was pulled back so that the IR camera could further record the load sequence.

The crosshead speed *v_c_* of the test machine was set up differently according to the strain rate for different phases during the same tensile test. Until the end of the yielding phase, it was fixed at 0.9 mm min^−1^. Then, until the final fracture, it was increased to 24 mm min^−1^.

Stress (force)-controlled constant amplitude tests (CAT), as well as load increase tests (LIT), were carried out uniaxially without superimposed mean stress (*R* = −1) at ambient temperature with a load frequency of 5 Hz and a sinusoidal load-time function on a servohydraulic testing system type EHF-U by Shimadzu (Shimadzu Europa GmbH, Duisburg, Germany). The complete experimental setup is shown in Figure 4.

To characterize the microstructure-based fatigue behavior in detail, the change in temperature Δ*T*, the change in MBN signal ratio Δ*φ_MBN_* and the change in electrical resistance ratio Δ*φ_R_* were calculated from the measurands. Technical specifications of the IR camera, the MBN sensor, as well as the preparation of the fatigue specimens, are already introduced in [37].

To ensure the acquisition of MBN signals, a sensor type µmagnetic by QASS (QASS GmbH, Wetter an der Ruhr, DE) was implemented into the test setup. The sensor head was mounted on an adapter made of copper fixed to a stable x/z-platform, in order to reach the test section in the middle of the vertically clamped fatigue specimen (Figure 4, marked with No. 2). 

The later extracted feature “change in MBN signal ratio” Δ*φ_MBN_* was derived from the peak value of the pre-processed MBN signals *I_max_* [37] with an arbitrary unit in the time domain. It is the mean value calculated from three cycles in the middle of the excitation window. After lift-off compensation, the MBN signal ratio was calculated by
Δ*φ_MBN_ = (I_max_ − I_max,0_)/I_max,0_*,(1)
where *I_max_* is the measured maximum value of the MBN signal intensity, and *I_max,0_* is the adjusted beginning value of each measurement.

To measure the electrical resistance of the tested specimens, a four-point principle measuring device was adapted, which consists of two wires attached to edges of the testing section on tensile/fatigue specimens, two electrodes on the shafts, one shunt-resistance, one DC power supply TOE8841-24 (Toellner Electronic Instrumente GmbH, Herdecke, DE) and a DAQ card type USB 2408 (Measurement Computing Corporation, Norton, MA, USA).

The other extracted feature “change in electrical resistance ratio” Δ*φ_R_* was calculated, as an analog of Δ*φ_MBN_*, from the electrical resistance R with a unit of [µOhm], which was measured during the load-free sequence shortly before the next load sequence started, where *R_0_* is the initial value of electrical resistance of each specimen in the origin state:Δ*φ_R_ = (R − R_0_)/R_0_*.(2)

### 2.3. MaRePLife: Principles

The MaRePLife (Material Response Partitioning for fatigue life estimation) is a modified short-time method based on StressLife [38]. The biggest difference to StressLife is the definition of the elastic and the plastic portion of the measurands. Additionally, the parameters of LIT and CAT are now clearly ruled based on tensile test results. Therefore, the first step here is always aimed at obtaining the *R_m_* and the *R_p0.2_* values (Figure 5a).

The LITs have the following functions: 1. To determine the elastic modulus of the continuously acquired physical quantities, which belongs to the “elastic material response partitioning” (in certain cases, this function can be replaced by a tensile test, as shown in Figure 6a). 2. To determine the stress amplitudes of both CATs to be conducted afterwards (Figure 5c). In StressLife, there was only one LIT conducted to define the cyclic strain hardening exponent *n’*. However, since the materials’ responses are quite different due to different heat treatments, the damage accumulation of one single LIT would lead to a very different *n’* value than that from separately conducted CATs, as Morrow showed in [39]. Otherwise, the so far fixed way of conducting load increase tests by applying stress amplitudes of (100 + 20*∙*n) MPa normally works well for steels regarding a starting stress amplitude of 100 MPa, but if a different material with a different yield and tensile strength is to be tested, this value can also be calculated generally by
*σ_a,start1_ = 0.25 ∙ R_p0.2_*.(3)

From the first LIT, two more characteristic mechanic features without much scattering for the same material can be obtained: the stress amplitude at which the first obvious increment of material response is observed, *σ_Y,f1_*, and the stress amplitude at which the specimen breaks, *σ_m,f1_*.

The only problem is that the application of (100 + 20*∙*n) MPa could result in a lack of data points in the plastic region, which strongly influences the quality of the regression calculation. Therefore, an additional second LIT is required to cover the elastic–plastic to mostly plastic region. The starting stress amplitude of the second LIT, *σ_a,start2_*, is defined according to the experimental results from the first LIT as
*σ_a,start2_ = 2 ∙ σ_Y,f1_ − σ_m,f1_*.(4)

Only load increase values smaller than 20 MPa (as for the first LIT) can cover the elastic–plastic region better. Therefore, a way of calculating the second load increase Δ*σ_a2_* is proposed by applying the following equation:Δ*σ_a2_* = *α_f_* ∙ Δ*σ_a1_ = ((σ_m,f1_ − σ_Y,f1_)/(R_m_−R_p0.2_)) ∙* Δ*σ_a1_*.(5)

The coefficient between both stress increments, α_f_, will be rounded with one single digit after the decimal separator, in order to obtain only even values for the Δ*σ_a2_*. *α_f_* is used to ensure that Δ*σ_a2_* would be proportionally quantified for different materials, so that the experimental duration can be kept rationally short. Table 3 shows the mechanical properties and parameters of LITs for the materials investigated within the frame of this work:

The NDT techniques involved in the short-time methods aim to track following three different physical quantities associated with fatigue processes: the temperature, captured by the IR camera; the electrical resistance, obtained by the voltage measuring DAQ module; and the magnetic Barkhausen noise, recorded by the micromagnetic sensor. While the first two physical quantities can be acquired continuously, the MBN signal is not quite able to be continuously recorded since the electromagnetic excitation is integrated in the sensor head. On the one hand, a long-time magnetization could result in overheating of the sensor. On the other hand, it is far too challenging for the data storage compacity of the testing system, especially when the duration of a fatigue test could become longer than expected.

In analogy to the definition of the elastic strain amplitude/range, which, for the elastic deformation, is exactly the same as the amplitude of the continuous periodic (sinusoidal) strain–time function, the elastic portion of the change in temperature or change in electrical resistance data *M_e_* can be exactly defined as the amplitude of the periodic signal (Figure 5b). According to the thermoelastic theory proposed by William Thomson (Lord Kelvin) [40], this quantity should be linearly correlated to the stress (amplitude). The net change during each load cycle, which is calculated here as the mean value of all data points from each load cycle, is considered as the plastic portion *M_p_*, even though the corresponding plastic strain amplitude/range is not exactly defined in the same way. However, practical experience has shown a very good correlation between both quantities [19,26,27] so that the authors find it reasonable to further apply it in this way. By observing the *σ_a_–M_p_* chart of both LITs, the stress amplitudes for both upcoming CATs can be determined. The first one should be in the transition range from the elastic to plastic region (Figure 5b and Figure 6b, the blue marked section), and the second one should mostly be in the plastic range (Figure 5b and Figure 6b, the red marked section).

After the CATs are conducted, the plastic portion of both tests *M_p,1_* and *M_p,2_* can be defined by calculating the mean value of the *M_p_–N* curve after the incubation period, i.e., after the first obvious cyclic softening (Figure 5c and Figure 6c). For further calculation, the application of the idea “material response partitioning” is essential (Figure 5d and Figure 6d). It shares a similar idea as StressLife, i.e., it is also based on the Coffin–Manson and Basquin laws [1,2,3]. This approach started with an analogy with the Ramberg–Osgood relationship [41] describing the non-linear relationship between stress and strain of monotonic tensile tests. The cyclic stress–strain curve was expressed mathematically after Morrow [39], in an equation which consists of an elastic part, described by Hooke’s law, and a plastic part which follows a power function. The strain range, Δ*ε*, in the manner of traditional fatigue test estimation, can then be described with the following equation: Δ*ε_t_* = Δ*ε_e_* + Δ*ε_p_ = σ_a,t_/E + ε’_f_ ∙ (σ_a,t_/σ’_f_) ^1/n’^*,(6)
where Δ*ε_t_*, Δ*ε_e_* and Δ*ε_p_* are the total, elastic and plastic strain range/amplitude, respectively; *σ_a,t_* is the total stress amplitude; *ε’_f_* and *σ’_f_* are constants related to the cyclic ductility and strength of the metal; and *n’* is the cyclic strain hardening exponent, which is also considered as a material-specific constant.

In order to reveal the relation of the measured material response (with regard to the conventional strain range) to the fatigue life, Morrow proposed an equation which integrates the Coffin–Manson relation and the Basquin equation [39], thus providing a direct link between the fatigue life *N_f_* (to be calculated) and the total strain amplitude Δ*ε_t_*, which is also sometimes referred to as the Coffin–Manson–Morrow equation:Δ*ε_t_ = (σ’_f_/E) ∙ (2N_f_)^b^ + ε’_f_ ∙ (2N_f_)^c^*,(7)
where *c* is called the fatigue ductility exponent and *b* the fatigue strength exponent.

By replacing the strain amplitude by the generalized material response *M* in Equation (7), each portion can be written as follows:Δ*M_t_* = Δ*M_e_* + Δ*M_p_*,(8)
Δ*M_e_* = *B ∙ (2N_f_)^b^*,(9)
Δ*M_p_* = *C ∙ (2N_f_)^c^*.(10)

In MaRePLife, for simplification, it is just assumed that the plastic portion of the measurands still contribute predominantly to the fatigue life. The calculation of the elastic portion could be performed in two ways, depending on the feature of the data acquisition.

The point of partitioning with a number of load cycles *N_pop_* and material responses of *M_pop_* can be solved by combining Equations (9) and (10):*N_pop_ = ((C/B)^1/(b – c)^)/2*,(11)
*M_pop_ = B ∙ ((C/B)^1/(b − c)^)^b^*.(12)

After the material response at the point of partition is known, it is easy to calculate the corresponding stress amplitude by multiplying the elastic modulus:*Σ_pop_ = E_M_ ∙ M_pop_*.(13)

Then, through a three-point allometric regression together with *M_p,1_* and *M_p,2_*, the cyclic strength coefficient *K’* and the cyclic strength exponent *n’* can also be solved. Noticing that the plastic portion of Equation (7) contains the term *σ_a,t_*_,_ and for stress-controlled CATs, *σ_a,t_ = σ_a_*, this results in
*σ_a_ = σ’_f,M_* ∙ *(*Δ*M_p_/ε’_f,M_)^n’^_M_ = K’ ∙ (C ∙ (2N_f_)^c^) ^n’^_M_*
(14)

Thus, the S–N curve is expressed with continuous *σ_a_* and *N_f_* values. The flowchart in Figure 7 summarizes this procedure with both cases shown in Figure 5 and Figure 6.

## 3. Results

### 3.1. Tensile Tests

At the beginning of each test series, a tensile test was carried out in order to simultaneously define the tensile strength *R_m_*, the yield strength *R_p0.2_*, as well as the elastic modulus *E_MBN_* of the MBN signal. It can be seen in Figure 8 that the temperature change during the loading sequences was becoming increasingly higher. This explains the necessity of conducting the MBN measurement discretely during strain-holding sequences where the engineering strain was kept constant, to avoid possible thermal impact. Since the tensile test is controlled by strain, the stress response during each holding sequence would drop slightly due to slight cold work hardening processes. The MBN measurements always took place at the end of each holding sequence, to make the MBN–stress correlation reliable. At the beginning of the test, it can be seen that the elastic response of the MBN signal is very sensitive and forms a grade line over the whole elastic region. After the yielding process, the MBN signal drops back and does not show a significant change as the plastic deformation keeps increasing, which corresponds to the result shown in [42].

By conducting linear regression for the stress to MBN signal ratio in the elastic region, the slope can be easily obtained, which is interpreted in this work as the elastic modulus of the MBN signal response *E_MBN_*. In Figure 9, all *E_MBN_* data of the materials investigated within the frame of this research project are listed. The data points are obtained until 0.2% strain.

It can be seen that the influence of heat treatment on both low carbon steels can be revealed clearly by *E_MBN_*: if the steel is strengthened and hardened after heat treatment, the MBN response will be higher, so the elastic modulus is higher. According to this value, the elastic portion of the MBN response during fatigue tests can be easily converted according to the corresponding stress amplitudes. It should be mentioned here that the regression calculation of all above-mentioned slopes is set to cross the (0, 0) point to maintain the physical validity, although the fit quality (R^2^) is locally sacrificed.

### 3.2. Load Increase Tests

The results of both LITs conducted on SAE 1020 (Figure 10) in normalized material conditions are shown below as an example for the operation of the MaRePLife procedure. All calculated plastic material responses derived from the three aforementioned measuring techniques are presented: the change in temperature, Δ*T*; the change in MBN signal ratio, Δ*φ_MBN_*; and the change in electrical resistance ratio, Δ*φ_R_*. As described in [37], the tests were not performed with continuous load, but with inserted load-free sequences. The progress of Δ*T_p_–N* curves are shown in a continuous way, while the mean values from each load step of Δ*T_p_*, Δ*φ_MBN,p_* and Δ*φ_R,p_* are emphasized with dots of different shapes. Obvious changes in the values of all three quantities in the plastic region can be observed. The second LIT had a connection problem in the ER measurement, so that the data points were unfortunately lost (from *σ*_a_ = 230 MPa onwards), but this does not change or disturb any further steps.

It can be seen in Figure 11 that the plastic portion of both materials generally increase, but at certain stage, the MBN signal falls, as already shown in [37]. The data point from the second LIT covered the elastic–plastic region and gives an idea of the choice of the stress amplitudes for the upcoming CATs. As mentioned before, the last four data points of the change in the electrical resistance ratio from the second LIT were lost.

To extract the elastic information, the amplitudes of the continuous signals measured during LITs were used, as mentioned in Section 2.3. Figure 12 shows zoomed-in *M–N* curves with an example of normalized SAE 1020, where the periodic Δ*T–N* and Δ*R–N* signals are broken down into amplitude and mean value for every single load cycle.

Figure 13 shows that the temperature data can especially result in an ideal fit quality at the linear regression, which is also well repeatable, though it is not the focus of this paper. By applying the rule of the zero y-axis intercept, i.e., the (0, 0) point should be on the fit line, Δ*φ_R_–N* data points also give good fit quality, but with a larger deviation.

Here, the elastic amplitude from each load step of the first LIT is usually chosen to determine the elastic modulus, since the starting stress amplitude begins at 100 MPa, which is definitely low enough to cause purely elastic deformation. For single cases of electrical resistance values, the second LIT could have a higher elastic modulus, the reason of which the reason is still not entirely clear. In this case, the higher value is chosen for the later calculation, regardless which LIT it is calculated from. Since the elastic amplitude is determined by calculating the half value of the difference between the maximum and the minimum from each single load cycle, the reliability of the elastic portion is only high in the real elastic region. Later, when plastic deformation appears, since the net change of the mean value is no longer zero, the half value of the difference would be higher than it actually is. As shown in Figure 13, after the first six–seven data points, the linearity is lost.

### 3.3. Constant Amplitude Tests

According to Figure 11, the first CAT should be conducted with a stress amplitude of 190–210 MPa, while the second one can be chosen as 224–240 MPa, which results in highly plastic responses during LIT. Depending on the data validity of single CATs, 204 and 230 MPa are chosen for the calculation with MBN data Δ*φ_MBN_* (Figure 14a). Δ*φ_R_* analysis was performed on CATs with 204 and 228 MPa as the stress amplitude (Figure 14b). Two more results for normalized SAE 5120 are shown in Figure 14c,d, where both amplitudes are chosen as 246 and 260 MPa according to the corresponding results of LITs.

On all four graphs in Figure 14, the data points and curves for the CAT with higher stress amplitude are given in orange and the other are in in blue. The discrete small data points are acquired from the load-free sequences. For describing the trend, a modified Langevin function is used for MBN data [37]. For the change in electrical resistance ratio, the Langevin function did not work well because of the value drop in the last period for ca. 1/6 *N_f_*. The plastic value for each CAT is calculated as the mean value after the incubation stage, shown as the big square points, which are located near to the 1/2 *N_f_* cycle.

### 3.4. MaRePLife

As shown in Figure 7, the calculation of *K’* and *n’* are essential to obtain the final S–N curve. If the *σ_a_–M_p_* curve can be validated by different CATs, then the quality of the calculated S–N curves will also be proved. In Figure 15, as an extension of Figure 11, *σ_a_–M_p_* curves of four different material conditions are demonstrated.

Either for MBN or for ER data, they are mostly located close to the calculated *σ_a_–M_p_* curves (dash-dotted lines); the deviation will be discussed later. The calculated values of *K’* and *n’* are marked on the graphs. It is interesting to see that some data points from the LITs in the plastic region can also be covered by this curve.

After *K’* and *n’* are solved, the S–N curves can be calculated. Figure 16 shows S–N curves derived from electrical resistance and MBN features for the selected materials/material conditions investigated within the frame of this work. It can be seen that for the HCF regime, the calculated S–N curves can be supported by the CAT data points. By selecting the parameters of CATs according to the results of tensile tests and load increase tests, the HCF regime of 10^4^–2 × 10^6^ can be described properly. Some of the measured values from the electrical resistance for SAE 5120 were invalid due to earlier experiments with outdated load-free sequence parameters.

## 4. Discussion

The result of the MBN signal at the tensile test, as shown in Figure 8, can be supported by the former work of [42], which reported a drop in the MBN signal (RMS and MAE, magneto-acoustic emission) after the plastic strain appears, i.e., after plastic deformation takes place. The present work also shows a strong drop in the MBN value at the beginning of plastic deformation but then tends to decrease gradually and inconspicuously until the final fracture (Figure 8), which is more like the trend reported in [43].

In Figure 9, the linearity between stress and the MBN response can be proved. Yet, there were values which fell below 0 at the beginning. The negative value is caused when the first two or three MBN signal intensity values in [a.u.] do not have big differences, so when the ratio is built according to Equation (1), it turns to be negative. Yet, it is observed to be trivial.

When comparing the material responses of both LITs, as shown in Figure 10 and Figure 11, it can be seen that the MBN signal reacts more sensitively to the accumulated damage (plastic deformation) in terms of the obviously elevated values, which is similar to the behavior of Δ*T* after *σ_Y,f1_*, while the ER signal only grows gradually. This phenomenon can also be confirmed with other materials investigated (Figure 15c,d). The ER signal is probably more sensitive to the macroscopic cracks while MBN is already sensitive to sub-microscopic defects due to its interaction with magnetic domain movement.

The trend curves shown in Figure 14 indicate a similar behavior of MBN and ER responses during CATs. However, the ER signals tend to have this characteristic drop in the last stage, which could again be caused by the formation of macroscopic cracks. The data acquisition by resistometry represents the voluminal information of the entire test section of the fatigue specimen, because the voltage drop is regarding the area between those two wires point-welded onto the specimen, while the MBN measurement only happens locally on one spot in the middle of the test section where the pick-up coil of MBN sensor is positioned. Therefore, it is possible that the effect of an existed crack might be missed by the MBN sensor, whereas the ER measurement can still catch it.

The approach of taking material response partitioning into power law Coffin–Manson–Morrow equation to calculate the S–N curve is derived from the StressLife method, which is also based on the “homologous” feature between plastic strain amplitude and the change in temperature caused by heat dissipation. The authors will show the results of MaRePLife in a future paper by using the change in temperature as input, which has a lower deviation. Since MBN and ER are not contactless methods, the quality of the data acquisition would affect the final effect of applying MaRePLife. Deviation, as shown in Figure 15, could eventually be eliminated if the experimental conditions can always be kept strictly the same.

The final results of the MaRePLife method in Figure 16 show good congruence between the calculated S–N curves and the single data points. Yet, the position-dependency on the selection of CAT parameters is also high. To obtain a better 50% P-S–N (probabilistic S–N) curve, statistical experiment design and analysis should be taken into consideration, which presents a meaningful research direction for the future.

## 5. Conclusions

This paper shows that NDT techniques regarding electrical resistance and micromagnetic measurements can provide useful features as input for short-time fatigue calculation methods such as MaRePLife. The conclusions are as follows:It is essential to perform tensile and load increase tests (LIT) in order to determine the parameters of constant amplitude tests (CAT), which is the key to conduct short-time fatigue calculations. A method depending on *R_m_*, *R_p0.2_* and the values from the first LIT (*σ_Y,f1_* and *σ_Y,m1_*) has been proposed, which is suitable for the common materials to be investigated.The elastic modulus of material responses is defined by calculating the slopes of *σ_a_–M_e_* data points in the elastic region from a single LIT or a tensile test. The latter is ideal for measurands acquired with discrete measuring methods, such as MBN.By applying the idea of material response partitioning for MBN and ER, we could maximize the information we acquired from single fatigue tests and use this as input to calculate the S–N curve in the HCF regime, so that the total cost of time can be reduced.

## Figures and Tables

**Figure 1 materials-16-00032-f001:**
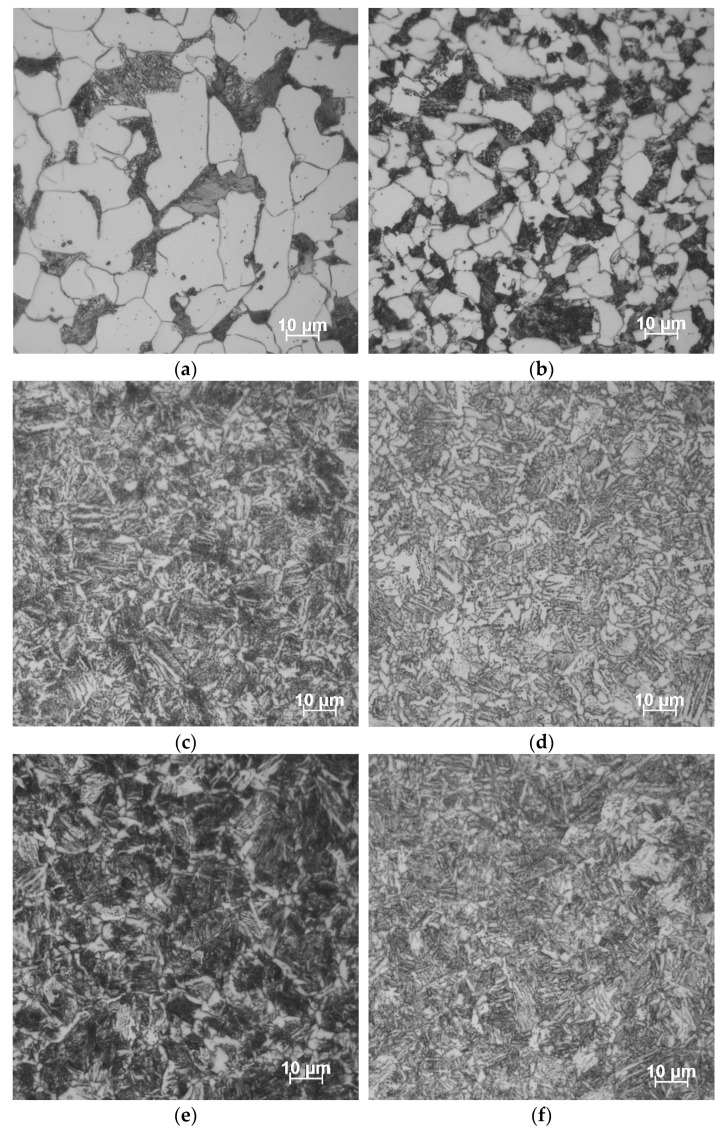
Micrographs of the investigated steels in different heat-treated conditions. SAE 1020: (**a**) +N; (**c**) +QT650; (**e**) +QT250. SAE 5120: (**b**) +N; (**d**) +QT650; (**f**) +QT250.

**Figure 2 materials-16-00032-f002:**
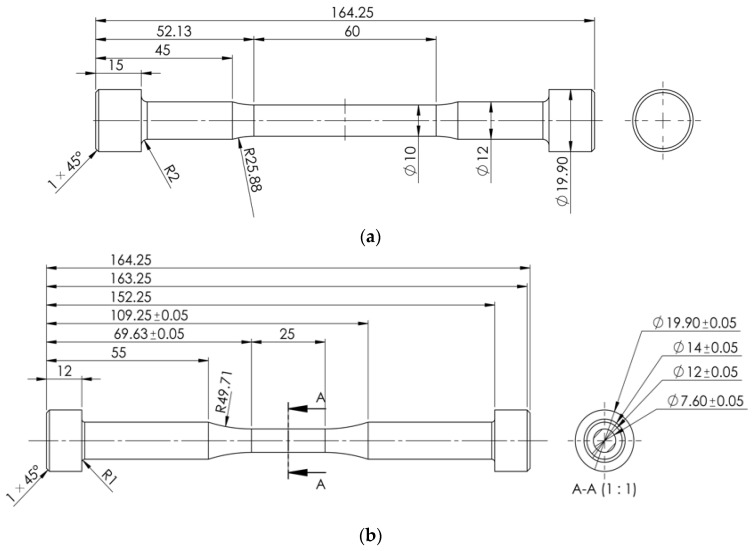
Dimensions of the applied (**a**) tensile specimen; (**b**) fatigue specimen.

**Figure 3 materials-16-00032-f003:**
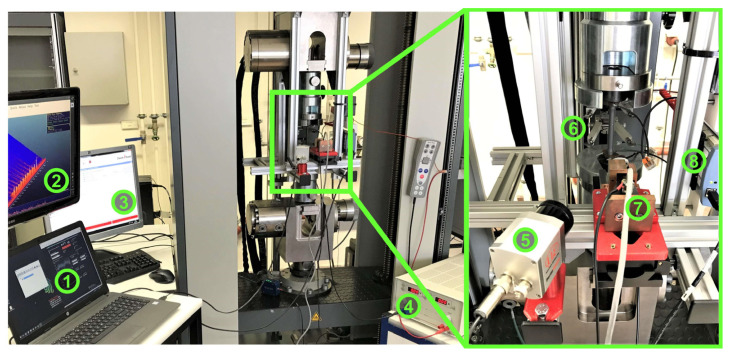
Applied tensile testing system with measuring techniques and related devices: 1. PC for DAQ (data acquisition) of the IR camera and ER; 2. PC for the MBN measurement; 3. PC for tensile test machine control; 4. DC power supply for ER; 5. IR camera; 6. Tactile extensometer; 7. MBN sensor mounted on a sliding rail; 8. Electrodes and sensing wires of the four-point DC resistance measurement with DAQ device.

**Figure 4 materials-16-00032-f004:**
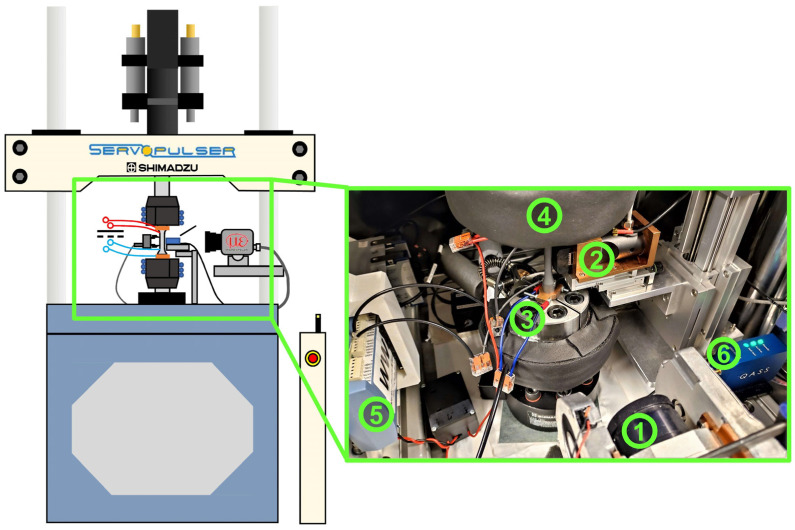
Applied fatigue testing system with measuring techniques and related devices: 1. IR camera; 2. MBN sensor; 3. Electrodes and sensing wires of the four-point DC resistance measurement device; 4. Cooled clamps; 5. DAQ device for electrical resistance measurement; 6. Pre-amplifier for MBN sensor.

**Figure 5 materials-16-00032-f005:**
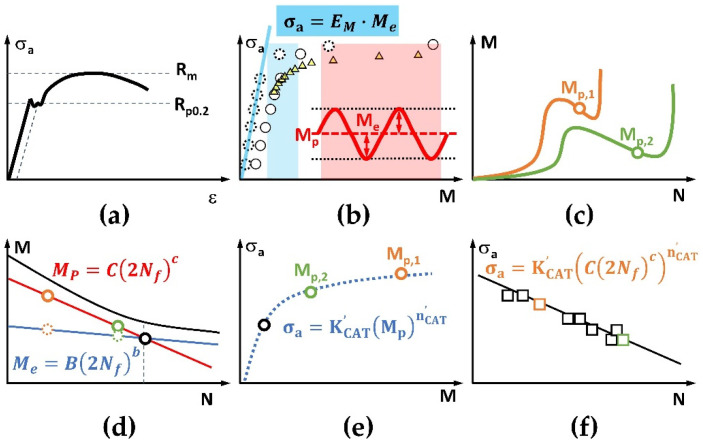
Schematic procedures of MaRePLife for measuring techniques with continuous data acquisition (e.g., thermometry or electrical resistometry). (**a**) Tensile test for *R_m_* and *R_p0.2_* determination. (**b**) First LIT (plastic portion: white round dots, solid line) and second LIT (plastic portion: yellow triangles) for the determination of the stress amplitude of both upcoming CATs (selected in the red and blue areas). The elastic response *M_e_* of the first LIT (white round dots, dotted line) for the determination of the elastic modulus (blue straight line). (**c**) Both CATs and their plastic response *M_p_*. (**d**) Elastic (round dots with dotted line) and plastic (round dots with solid line) portion calculated following Basquin and Coffin–Manson laws to find the partitioning point (round dot, black solid line). (**e**) Allometric regression of the plastic responses for obtaining *K’_CAT_* and *n’_CAT_*. (**f**) Calculated S–N curve with both CATs used and more CATs for verification.

**Figure 6 materials-16-00032-f006:**
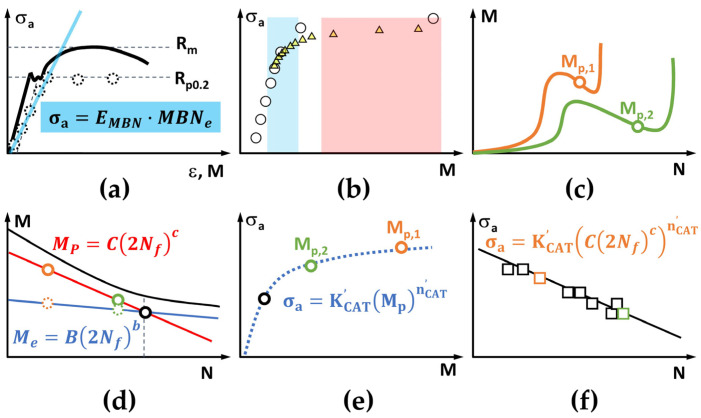
Schematic procedures of MaRePLife for measuring techniques with discrete data acquisition (e.g., magnetic Barkhausen noise). (**a**) Tensile test for *R_m_* and *R_p0.2_* determination. The slope (blue straight line) of the material response (round dots with dotted line) in the elastic region is used for the determination of the elastic modulus. (**b**) First LIT (plastic portion: white round dots, solid line) and second LIT (plastic portion: yellow triangles) for the determination of the stress amplitude of both upcoming CATs (selected in the red and blue areas); (**c**) Both CATs and their plastic response *M_p_*. (**d**) Elastic (round dots with dotted line) and plastic (round dots with solid line) portion calculated following Basquin and Coffin–Manson laws to find the partitioning point (round dot, black solid line). (**e**) Allometric regression of the plastic responses for obtaining *K’_CAT_* and *n’_CAT_*. (**f**) Calculated S–N curve with both CATs used and more CATs for verification.

**Figure 7 materials-16-00032-f007:**
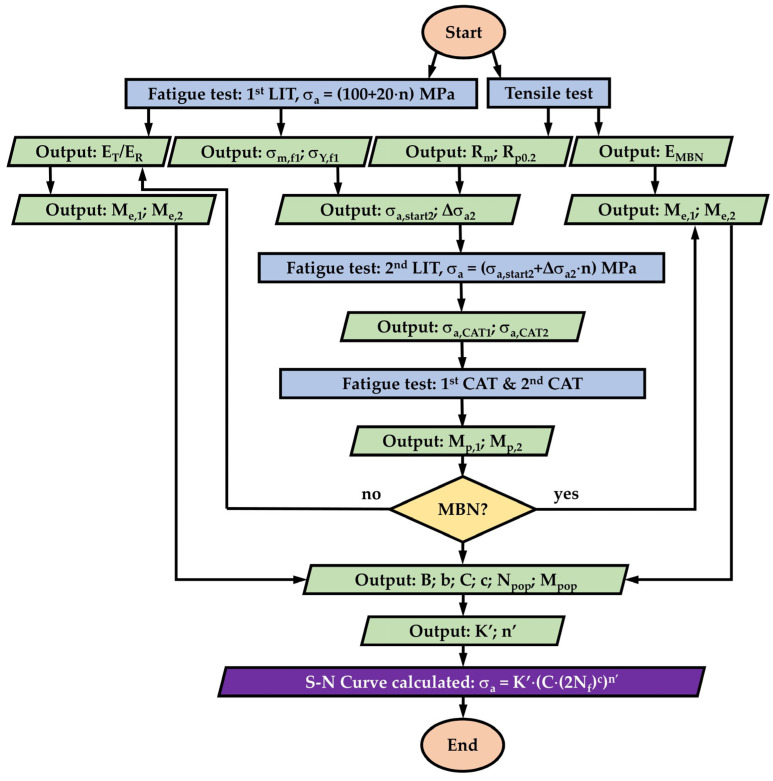
Flowchart of obtaining the S–N curve according to the principles described in Figure 5 and Figure 6.

**Figure 8 materials-16-00032-f008:**
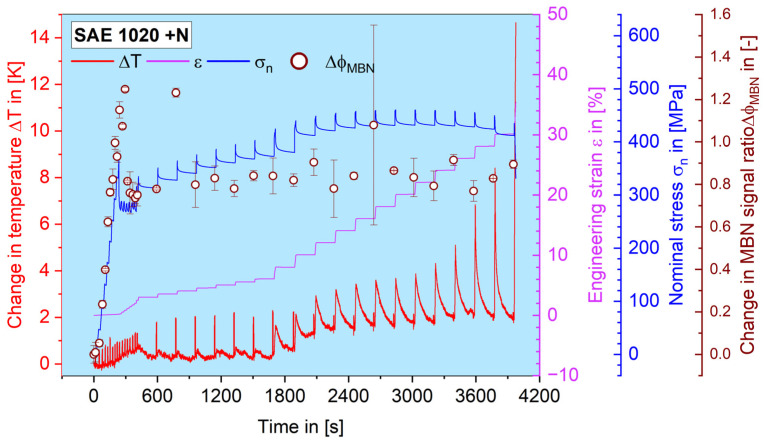
Results of tensile test on SAE 1020 steel in normalized conditions. The change in temperature, the engineering strain, the nominal stress and the change in MBN signal rate are shown as material responses.

**Figure 9 materials-16-00032-f009:**
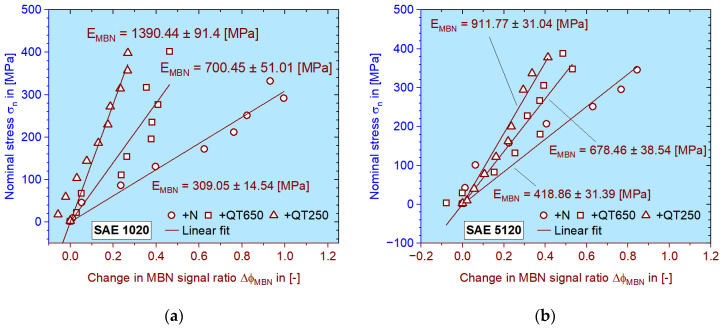
MBN signal responses in terms of Δ*φ_MBN_* caused by elastic deformation during tensile tests on (**a**) SAE 1020 and (**b**) SAE 5120 in the differently heat-treated (+N: normalized, +QT650: quenched and tempered at 650 °C and +QT250: quenched and tempered at 250 °C) conditions. The slopes are interpreted as the elastic modulus for the MBN signal *E_MBN_*, which has MPa as its unit.

**Figure 10 materials-16-00032-f010:**
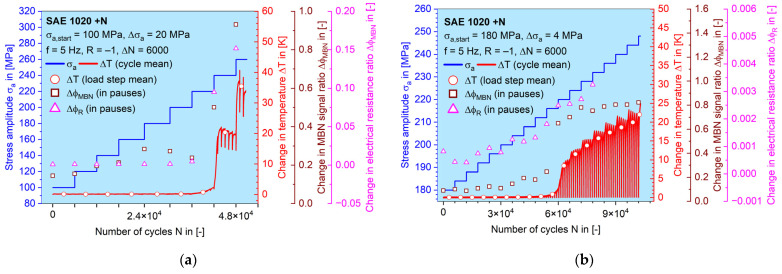
Development of the measured and calculated quantities plotted against the number of load cycles during (**a**) the first LIT and (**b**) the second LIT on SAE 1020 in normalized conditions. Mean values from each load step are shown as dots with different shapes.

**Figure 11 materials-16-00032-f011:**
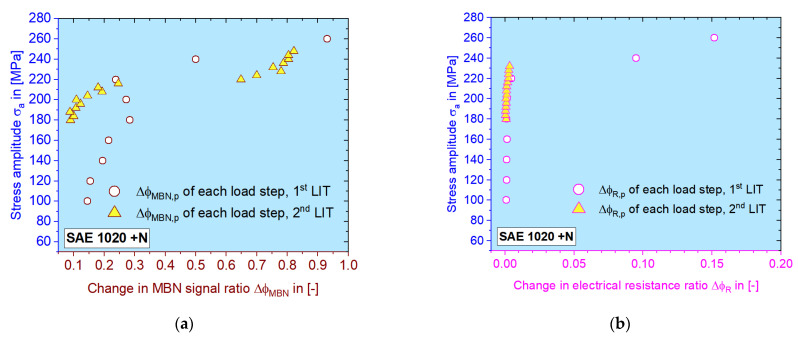
Mean plastic material response from each load step of (**a**) Δ*φ_MBN_* and (**b**) Δ*φ_R_* during both LITs, and their relation to the applied stress amplitudes for normalized SAE 1020.

**Figure 12 materials-16-00032-f012:**
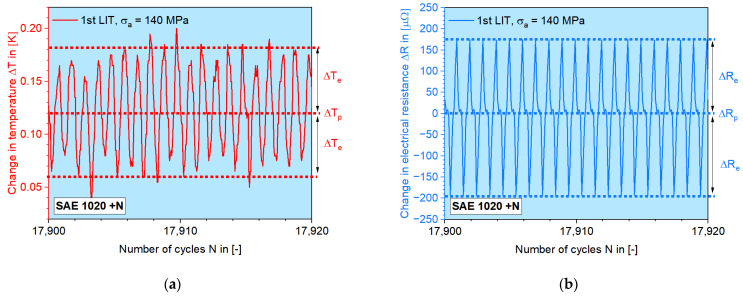
Cyclic material response in terms of (a) the change in temperature Δ*T* and (b) the change in electrical resistance Δ*R*. The amplitude in the elastic region is used to calculate the elastic portion. The mean value of each load cycle is calculated as the plastic portion.

**Figure 13 materials-16-00032-f013:**
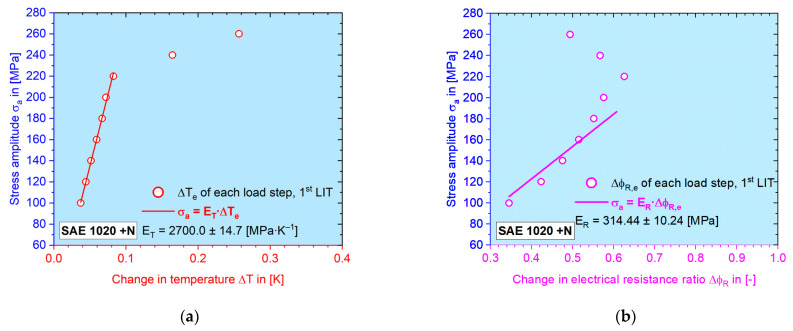
Calculation of the elastic modulus with the example of normalized SAE 1020 for (**a**) the change in temperature, Δ*T* and (**b**) the change in the electrical resistance ratio, Δ*φ_R_*, by performing linear regression on *σ_a_–M_e_* data points in the elastic zone.

**Figure 14 materials-16-00032-f014:**
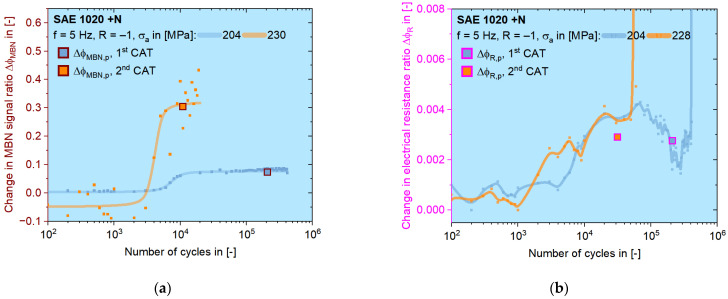
Development of single data points with trend curves during CATs on SAE 1020 of (**a**) Δ*φ_MBN_* and (**b**) Δ*φ_R_*, as well as on SAE 5120 of (**c**) Δ*φ_MBN_* and (**d**) Δ*φ_R_*. Both plastic values calculated from CATs (big square points) are later used for the fatigue life calculation.

**Figure 15 materials-16-00032-f015:**
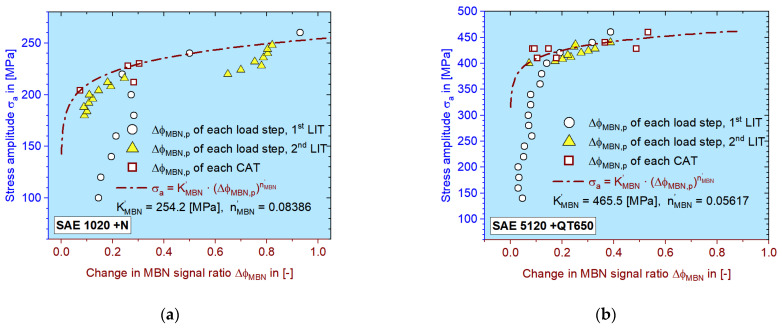
Stress amplitude of CATs as a function of plastic material response for the case of: (**a**) SAE 1020 +N; (**b**) SAE 5120 +QT650; (**c**) SAE 1020 +QT650; (**d**) SAE 5120 +N. The square data points represent the plastic portion from single CATs. The dash-dotted curves represent the calculated *σ_a_–M_p_* relation.

**Figure 16 materials-16-00032-f016:**
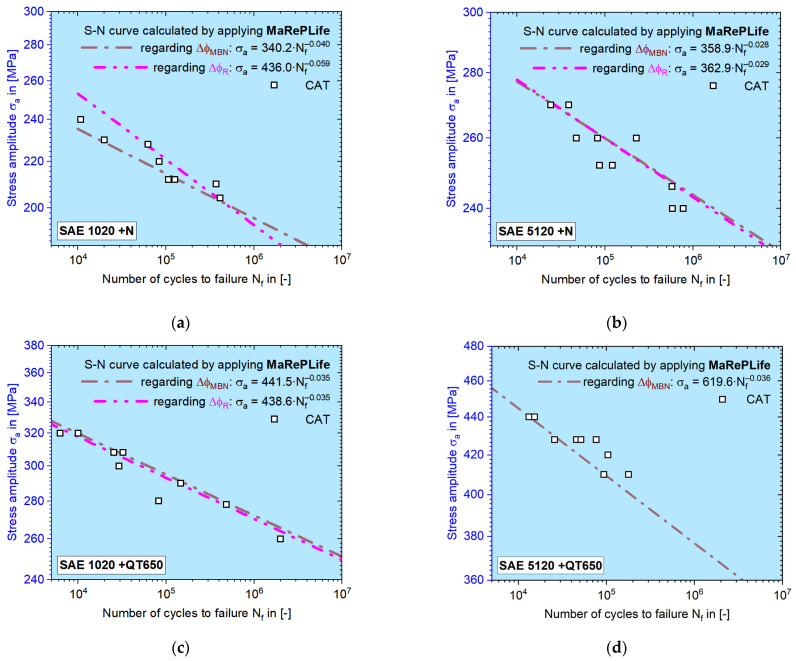
Calculated S–N curves using MaRePLife for (**a**) SAE 1020 normalized; (**b**) SAE 5120 normalized; (**c**) SAE 1020 quenched and tempered at 650 °C; (**d**) SAE 5120 quenched and tempered at 650 °C. The magenta dash-dot-dot-dash line is the S–N curve calculated by applying a change in the electrical resistance ratio, Δ*φ_R_*. The brown dash-dot-dash line represents the S–N curve calculated by applying a change in the MBN signal ratio, Δ*φ_MBN_*.

**Table 1 materials-16-00032-t001:** Nomenclature list.

Symbol or Abbreviation	Meaning	Symbol or Abbreviation	Meaning
CAT	Constant amplitude test	*E*	E-modulus
DAQ	Data acquisition	*I_max_*	Maximum MBN signal intensity
ER	Electrical resistometry	*I_max,0_*	Initial maximum MBN signal intensity
HCF	High cycle fatigue	*K’*	Cyclic strength coefficient
IR	Infrared	*L_0_*	Measuring length at tensile specimen
LCF	Low cycle fatigue	*M_e_(M_e,1_/M_e,2_)*	Elastic material response (of first/second CAT)
LIT	Load increase test	*M_p_(M_p,1_/M_p,2_)*	Plastic material response (of first/second CAT)
MaRePLife	Material response partitioning fatigue life evaluation	*M_pop_*	Material response at point of partitioning
MBN	Magnetic Barkhausen noise	*N_f_*	Fatigue life/number of cycles to failure
NDT	Non-destructive testing	*N_pop_*	Number of load cycle at point of partitioning
+N	Normalized state	*n’*	Cyclic strain hardening exponent
+QT	Quenched and tempered state	*R_0_*	Initial electrical resistance
*α_f_*	Ratio between the stress increases of both LITs	*R_m_*	Tensile strength
*b*	Fatigue strength exponent	*R_p0.2_*	Yield strength
*c*	Fatigue ductility exponent	*σ’f*	Cyclic strength
Δ*ε_t_/*Δ*ε_e_/*Δ*ε_p_*	Change in total/elastic/plastic strain	*T_aust._*	Austenitization temperature
Δ*φ_MBN_*	Change in MBN signal ratio	*T_temp._*	Tempering temperature
Δ*φ_R_*	Change in ER ratio	*v_c_*	Crosshead speed
Δ*σ_a1_*	Stress increase of the first LIT	*σ_a_/σ_a,t_*	(Total) stress amplitude
Δ*σ_a2_*	Stress increase of the second LIT	*σ_a,start1_*	Initial stress amplitude of the first LIT
Δ*T*	Change in temperature	*σ_a,start2_*	Initial stress amplitude of the second LIT
*ε _a,t_/ε _a,e_/ε_a,p_*	Total/elastic/plastic strain amplitude	*σ_m,f1_*	Stress amplitude at which the specimen breaks during LIT
*ε’_f_*	Cyclic ductility	*σ_Y,f1_*	Stress amplitude at which the first obvious increment of material response is observed during LIT
*E_M_/E_MBN_/E_R_*	E-modulus regarding material response/MBN/R		

**Table 2 materials-16-00032-t002:** Chemical composition in wt.-% of SAE 1020 and SAE 5120 tested: values according to DIN standards as benchmarks for producer’s information and our own analysis.

Material		Fe	C	Si	Mn	P	S	Cr	Mo	Ni
1.1149 SAE 1020	DIN EN 10083-2	-	0.17–0.24	≤0.4	0.40–0.70	≤0.030	0.020–0.040	≤0.40	≤0.10	≤0.40
	Producer	-	0.21	0.24	0.46	0.013	0.023	0.12	0.013	0.12
	* +N	bal.	0.234	0.286	0.481	0.014	0.018	0.118	0.014	0.112
	* +QT650	bal.	0.237	0.288	0.478	0.014	0.019	0.118	0.014	0.114
	* +QT250	bal.	0.236	0.286	0.479	0.014	0.019	0.119	0.016	0.115
1.7149 SAE 5120	DIN EN 10084	-	0.17–0.22	≤0.4	1.10–1.40	≤0.025	0.020–0.040	1.00–1.30	-	-
	Producer	-	0.18	0.24	1.23	0.015	0.026	1.05	0.022	0.10
	* +N	bal.	0.196	0.274	1.324	0.017	0.023	1.065	0.025	0.093
	* +QT650	bal.	0.194	0.274	1.309	0.016	0.024	1.056	0.022	0.093
	* +QT250	bal.	0.195	0.271	1.310	0.016	0.024	1.060	0.024	0.094

* Analysis of own samples.

**Table 3 materials-16-00032-t003:** Mechanical features of SAE 1020 and SAE 5120 tested.

Material Number after EN/SAE	Heat Treatment	*R_m_* [MPa]	$R_{p0.2}$ [MPa]	$\sigma_{m,f1}$ [MPa]	$\sigma_{Y,f1}$ [MPa]	$\alpha_{f}$ [-]	$\sigma_{a,start2}$ [MPa]	$\Delta \sigma_{a2}$ [MPa]
1.1149SAE 1020	+N	460	290	260	220	0.2	180	4
	+QT650	580	440	340	300	0.3	260	6
	+QT250	470	595	360	300	0.5	240	10
1.7149SAE 5120	+N	540	340	300	240	0.3	180	6
	+QT650	700	600	460	440	0.2	420	4
	+QT250	1200	905	760	580	0.6	400	12

## Data Availability

Not applicable.

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
