# Peer review of "Short-Time Fatigue Life Estimation for Heat Treated Low Carbon Steels by Applying Electrical Resistance and Magnetic Barkhausen Noise"

_materials, 2022, doi:10.3390/ma16010032_

Round 1

Reviewer 1 Report

There are many technique of stress measurements, the Barkhausen noise is one of the relatively new not very common. Of course relatively new  means more than 20 years old, however this technique is still developed and among other  not many producer of researches devices use it. Like other in some cases is better in other worst. For testing of steels, magnetic grades this method is very useful.
As for the Short-time fatigue life tests is also one of the technique used in many type of researches where knowledge of stress values is an information good enough.
When we take into account this paper the authors in my opinion have placed all needed information. Of course some minor faults can be found, however the paper as overall is even now is suitable for publication.
Taking into account above I sustain my opinion that the paper in this scientific form is suitable for publication.

Author Response

Thank you for your opinions. If it’s remarked as “can be improved” in every option, it would be good if we could get hints on where exactly to improve. We have revised the paper again and hope that we meet your expectations. Changes have been added and marked in red color.

Reviewer 2 Report

This paper combines tensile and fatigue test with measured ER and MBN responses, which helps to give S-N curves rapidly. It is interesting and meaningful for the fatigue issue on steels. However, according to the reviewer, the following major questions/suggestions should be explained/justified:

1.      As there are a lot of abbreviations in the text and in the figures, it is recommended to add a nomenclature list in the beginning.

2.      Line 16: Is “StressLife”, with such a form, referring to a specific method? It is unnecessary to use such a term in case the stress-life relation, i.e. S-N curves, is referred to.

3.      Eq. (3): Is the constant of 0.25 determined following a rule of thumb? Or is there any references can be provided?

4.      Line 205: Just to be clear, is M_p given by the response of electric resistance? If so, the MBN responses do not involve in providing the S-N curves. In another word, it seems that the MBN and ER can be used separately, as shown by Fig. 5 and Fig. 6, respectively. However, it is not exactly the procedure given by Fig. 7.

5.      Fig. 5: How is the point M_p,1 and M_p,2 in (c) determined for giving the red line in (d)? Is it more reasonable to use the value at the turning point?

6.      Fig. 9: The way of determining the upper and lower limit should be addressed for the sake of the data discretization. The same applies to Fig. 13, too. Also, some data points seemed erroneous by giving negative MBN change. Discussions should be given on that.

7.      Overall, it is not quite clear to me how is the procedure in Fig. 7 helps to shorten the acquisition of S-N curves. The two CATs can surely present a S-N curve (in the HCF regime, as mentioned in the paper), while the uncertainties in fatigue tests still requires more tests. As shown in Fig. 16, several fatigue tests with different stress ranges were carried out. Above all, by measuring the MBN and ER during the test, the change of strain can be considered, which seems beneficial to the accuracy of the fatigue strength assessment, but not the efficiency. Anyway, I believe it is worth further illustrations and explanations. And it could be another aspect to be mentioned in the conclusion part.

Author Response

Thanks a lot for your objective, constructive and extensive opinions! Changes have been added and marked with red color.

  1. done.
  2. The StressLife metioned here is, I have to admit, a little bit confusing. It’s not the famous established method stress-life, but as later in introduction with reference [38] shown, a method which belongs to the short time methods and was proposed by Prof. Starke.
  3. Yes, it’s only a rule of thumb according to our experiences, but quite reasonable, since the elastic linearity can be assured with this stress amplitude.
  4. Mp is just the generalized plastic material response, therefore the letter M. It can generally represent the physical quantity extracted from an NDT method. E.g. T for the case of temperature measurement. So MBN and ER are indeed different in our case, since the later one is acquired continuously and contains the information extracted during fatigue loading. But it only affects the calculation for Me. That’s the reason I describe this method with Figure 5 (suitable for ER) and 6 (suitable for MBN) separately. The Value of Mp would be calculated similarly. In Figure 7, the Difference between MBN and ER can also be recognized by following the decision step (yellow), if it’s MBN or not. If yes, then Me will go right and be calculated as Figure 6 shown. If not (as for Temperature and ER), then it goes left and follows the principle of Figure 5 to calculate Me. I hope my explanation helps.
  5. as written in line 224-225, the both Mp values are calculated as the mean value of Mp-N curve after the incubation period, which is near to the value we used to extract at N_f/2. The red line in (d) follows the equation written in red directly above it, so I haven’t really understood, what do you mean with “to use the value at the turning point”?
  6. For Fig. 9 we take the boundary where the 0.2% of strain is reached. For Figure 13 it is just  where the linearity obviously ends. Negative value comes when the first 2 or 3 values of MBN signal intensity in [a.u.] don’t have big differences, so when the ratio is built according to Eq. (1) it turns to negative. (add this to the revised version)
  7. Figure 7 is actually only a combination of Figure 5 and 6 but in form of a flowchart. As I mentioned in discussion (line 429-432), we are indeed considering to introduce statistical analysis into this method as it is still to be improved. The measurement of strain is considered to be a conventional way to assess the fatigue progress for sure. We do also measure it with a tactile extensometer. And in another published papers e.g. [27] of us, we already showed the congruence between plastic strain amplitude and the change in temperature. The purpose of showing the other data points was to verify this method, as a part of the project. It was not the focus of this recent project to find a perfect reliable way of getting S-N curve but just to explore the possibility of calculating S-N curve with electrical and micromagnetic quantities similarly as we did with the temperature. Afterall, with a specimen, we could measure as many quantities as we want. But in practice, the measurement of strain especially for a component could be very limited possible for certain situations.

Reviewer 3 Report

The short-time fatigue life estimation for heat treated low carbon steels is focused on in this paper, through conducting tensile tests and fatigue tests on differently heat-treated low carbon (non- and low-alloy) steels, and combining the non-destructive electrical resistometric (ER) and magnetic Barkhausen noise (MBN) techniques, aiming to establish an improved short-time fatigue life estimation method according to StressLife. The concept proposed in this paper is really interesting, and the results gotten in the paper seem to be useful for the further development of fatigue technologies. While, there are still some doubts need to be solved, to make the paper more innovative and practical:

1. The “MaRePLife” proposed in this paper is interesting, while how to illustrate the applicability of this concept to traditional HCF, or relationship between them? Is this concept or method can only used for HCF, how about LCF?

2. The critical method used in this paper is the non-destructive testing (NDT). However, as the reviewer know, the technology of non-destructive testing is currently very limited, especially the temperature-related measurement results mentioned in the manuscript (like line 189, “captured by the IR-camera......”). Actually, it may have a large error with the real situations. So, how does the author consider these problems? Is there any effective solution? If the data needed here can’t be detected precisely, the method proposed may be not useful for the real applications.

3. As the author said: “Therefore, it is possible that the effect of an existed crack might be missed by the MBN sensor, while the ER measurement can still catch it.” and “Since MBN and ER are not contactless method, the quality of the data acquisition would affect the final effect of applying MaRePLife. Deviation as shown in Figure 15 could eventually be eliminated if the experimental conditions can always be kept strictly the same.”......

 According to the reviewer's understanding, the author adopted many assumptions in the study, so how to explain the accuracy of the test results and the feasibility of the proposed concept?

4. Some minor typos need to be checked, like “link between the to be......” in like 242 and other places.

A very interesting concept is proposed in this paper. Nevertheless, the authors are required to answer the above questions clearly, and make detailed modifications to the manuscript. Then, the paper can be considered to be accepted.

Author Response

Thank you for bring in more discussions and help us to improve our paper.

Changes have been added and marked in red color.

  1. Well we now focus only on HCF, since the experiments were conducted force(stress)-controlled. For LCF region, it could also be interesting, yet if we change our testing program to strain-controlled tests, the feasibility of the equations we used here could be problematic. The cyclic deformation curves would also look differently. For this reason, we haven’t conducted constant amplitude tests with extremely high stress amplitudes, so that the fatigue life of none of our specimens was obviously below 104 cycles.
  2. That is a good point, since no matter what we measure, the precision and the repeatability of the chosen technique are always essential for delivering reasonable results. As far as we know, the IR-technology has already been applied quite well in this field, started already in the 1970s with TSA (thermal-elastic stress analysis). Researchers in Canada, the USA, China and especially France are still doing researches in this field regarding the self-dissipated heat during mechanical loading. Actually, our method is also tested by the temperature measurement, which would be published soon. So, we are actually optimistic towards this method. Otherwise, with techniques like ER or MBN, we could only try to standardized the way of specimen preparation and to rule the way of mounting our sensor head and try to fix those parameters. As another reviewer also asked why we don’t just measure the strain to increase the reliability, I’ve answered that there are always application scenarios where one measuring technique is not suitable. We hoped we could have a larger tool-box.

While the DIC technique which measures the strain field contactless was and is still under questioning sometimes, it has been improved continuously and we can see a bright future of this technique. So, we hope that it would be similar for MBN and ER too.

  1. OK, we have to come up with certain assumptions, which are considered being closed to the experiences from our tests. The test specimens have a cylindric test section with a length of 25 mm in the middle and a diameter of 7.6 mm, it is hard to extract the complete voluminal information with any single measuring technique. With thermography or DIC, which deliver 2D + intensity information, we could still only get the information of one side of our specimens. Let alone the MBN sensor has only one small pick-up coil. Even if with a conventional tactile extensometer, we can only clamp it on one side of the specimen to measure the local strain. So, I think the essential point of integrating NDT techniques into this proposed concept is not to cover the existed data 100%, but rather if the extracted information is correlated with the physical processes happening during fatigue tests. If there’s good correlation between them, then it is already applicable.

Regarding the quality of the obtained data, there are many factors which could have an influence on the results: temperature, surface roughness, geometric repeatability and the sensor head mounting etc. We could only eliminate part of them, but not everything. Our experiments lasted months long and were conducted both in winter and summer. As mentioned in the manuscript, we applied a different duration of the load free sequence at the beginning of the project. That’s what I mean with “if the experimental conditions can be kept strictly the same”. As the Figure 15 showed, the most single data points of the validation experiments (constant amplitude tests, CAT) are in the near of the calculated sigma_a-M_p curve but with deviation. So the accuracy seems not bad at this point, but we could still try to increase the repeatability of our results, so that the overall quality can be improved too.

We could also conduct more tests on the same stress amplitude, so that a statistical distribution could make the results more representative.

  1. thank you.

Round 2

Reviewer 2 Report

The authors have answered most of the questions in a proper way. Two more problems should be further explained before acceptance:

1.      Reply to Q5: it should be clearly defined what is the “incubation period”, as the Mp-N curves in Fig. 5c and Fig. 6c could be divided into different stages with respect to the obvious ups and downs. By the way, the “turning point” I mentioned is referring to the inflection point on the Mp-N curve, after which Mp increases sharply.

2.      Reply to Q7: According to the authors’ reply, I tend to stress the proposed method, by using ER and MBN measurements, on the advantages of feasibility and applicability for practical concerns, instead of being a “short-time” way of fatigue life estimation.

Author Response

Thanks a lot for your feed-backs regarding our answers. The changes we did now regarding the review report (round 2) are marked with green color in the manuscript.

1. 

We added in line 228 following description “[…], i.e. after the first obvious cyclic softening” to make it clearer.

       And thank you for suggesting us to use the “turning point”. The reason why we don’t use the value around this point is, that the cyclic hardening is usually not so strong for every material. Some authors even thought the stage after the first obvious cyclic softening as totally constant in their models. If we use the turning point shortly before the last sharp increase, then this progress would be more likely controlled by the macro crack formation but not only the material response. Therefore, to make it more characteristic, we choose to calculate the mean value after the incubation period, which could cover the material response quite well, no matter if the cyclic hardening is very strong or not. The schematic illustration in Figure 5c and 6c looks a little bit exaggerating with the cyclic hardening, but it is still possible especially for the case of normalized steels.

2. 

We are using ER and MBN beside the measuring of Temperature and Strain, in order to make the most of one single fatigue test. It is not that the usage of ER or MBN could be directly seen as a “short-time” way of fatigue life estimation which offers the reduction of experimental effort than other methods, but that the extra information we get with ER and MBN enriches the fatigue life estimation, so that less experiments in total will be conducted. In this sense, we could still regard the whole framework as a short-time approach.

       We rephrased therefore the last point of our conclusion as “3. By applying the idea of material response partitioning for MBN and ER, we could maximize the information we acquired from single fatigue tests and use this as input to calculate S-N curve in the HCF regime, so that the total cost of time can be reduced.”

And thanks again for your very frank, helpful and constructive opinions, we really enjoyed it :)